# Effects of Hand-Washing Facilities with Water and Soap on Diarrhea Incidence among Children under Five Years in Lao People’s Democratic Republic: A Cross-Sectional Study

**DOI:** 10.3390/ijerph18020687

**Published:** 2021-01-14

**Authors:** Yuko Noguchi, Daisuke Nonaka, Sengchanh Kounnavong, Jun Kobayashi

**Affiliations:** 1Graduate School of Health Sciences, University of the Ryukyus, 207 Uehara, Nishihara-cho, Nakagami-gun, Okinawa 903-0125, Japan; marubastusankakushikaku@yahoo.co.jp (Y.N.); junkobalao@gmail.com (J.K.); 2Lao Tropical and Public Health Institute, Ministry of Health, Samsenthai Road, Ban Kaognot, Sisattanack District, Vientiane 01030, Laos; sengchanhkounnavong@hotmail.com

**Keywords:** children, hand-washing, soap, diarrhea, household, Laos

## Abstract

Diarrhea is a leading cause of death among children under five (U5) in Lao People’s Democratic Republic (PDR). This study assessed the association between the presence of household hand-washing facilities with water and soap and diarrhea episodes among children U5 in Lao PDR. Data from the Lao Social Indicator Survey II were used. The outcome variable was diarrhea episodes in the two weeks preceding the survey. The main predictor variable was the presence of household hand-washing facilities with or without water and/or soap. Mixed-effect logistic regression analysis was used to assess the association, controlling for clustering, and other predictor variables. Of the 8640 households surveyed with 11,404 children, 49.1% possessed hand-washing facilities with both water and soap and 34.7% possessed hand-washing facilities with water alone. Children whose households possessed hand-washing facilities with water alone were significantly more likely to have a diarrhea episode compared to children whose households possessed hand-washing facilities with both water and soap (8.1% vs. 5.9%; odds ratio, 1.49; 95% confidence interval, 1.22–1.81). The association remained significant even after adjusting for other predictors. The absence of soap in hand-washing facilities was associated with higher odds of having a diarrhea episode among children U5 in Lao PDR.

## 1. Introduction

Diarrhea is a leading cause of death among children under five years of age (U5), especially in low- and lower middle-income countries [1,2]. According to the World Health Organization, approximately 525,000 children U5 die from diarrheal disease each year [3]. Lao People’s Democratic Republic (PDR) is a lower middle-income country located in Southeast Asia. In Lao PDR, the mortality rate of children U5 remains high compared to those in neighboring countries such as Thailand, Vietnam, and Cambodia, which is due partly to diarrhea [4,5,6]. The Global Burden of Diseases Diarrhoeal Diseases Collaborators estimated that the mortality of diarrheal disease in children U5 was 97.1/100,000, and approximately 1,000,000 episodes occurred in children U5 in Lao PDR in 2015 [7].

Hand-washing in everyday life is effective for reducing the risk of diarrhea. A systematic review examined the impact of hand-washing with soap on the risk of diarrheal disease and reported that hand-washing with soap in community settings can reduce the risk of diarrheal disease by 42–47%; therefore, interventions to promote hand-washing can save millions of lives a year [8]. According to another systematic review that assessed the effects of hand-washing promotion interventions on childhood diarrhea, community-based hand-washing promotion in low- and middle-income countries can reduce the incidence of diarrhea in children by approximately a quarter [9].

Rapid observations are widely used as a proxy measure of hand-washing practice, assuming that household members practice hand-washing with soap if a specific place for hand-washing is observed in a household with available water and soap [10]. In a nationwide survey, such as the Multiple Indicator Cluster Survey and the Demographic and Health Survey, rapid observations are widely adopted, since it can be challenging to measure hand-washing practice by direct observation because of the enormous associated costs and times. The Lao Social Indicator Survey II (LSIS II), a household-based nationwide survey that was implemented by the Lao government in 2017, also used rapid observations.

In Lao PDR, expanding the coverage of water, sanitation, and hygiene (WASH) facilities has been prioritized especially in rural areas and many WASH projects including open defecation free projects have been implemented [11]. However, very few projects measured project’s impact on health outcomes including diarrhea. A comprehensive WASH in school project, which included water supply, sanitation, hand-washing, and behavior change interventions, was implemented in 492 primary schools across 13 provinces between 2013 and 2017 [12]. A cluster randomized controlled trial study was done in Saravan province to measure the project’s impact on pupil absence, diarrhea, respiratory infection, and soil-transmitted helminth infection. The study found, however, that even among schools with the high level of fidelity and adherence, impact of the intervention was minimal. Thus, the study concluded that WASH in school alone may not achieve significant health gains without concurrent community and household WASH improvements including the improvement of hand-washing [13].

The association between hand-washing facilities with soap and diarrhea incidence among children U5 remains poorly understood in Lao PDR. To the best of our knowledge, two studies have assessed this association in Lao PDR. A cross-sectional study with 297 households in 50 villages in Saravan province examined associations between the presence of household hand-washing facilities and the infection status of diarrhea-causing pathogens among household members, including children U5 [14]. The study found that the presence of household hand-washing facilities was associated with lower infection rates of viral enteric pathogens and soil-transmitted helminths among household members. Another longitudinal study assessed diarrheal risk factors with 234 households from two villages in Saravan province, including all household members [15]. The study found that the presence of hand-washing with soap near or in the toilet was not associated with self-reported diarrhea episodes. The LSIS II included both rapid observation of the hand-washing facilities with soap and diarrhea episodes in children U5 [4]. However, no analysis has been made on the association.

Therefore, the present study aimed to assess the association between the presence of household hand-washing facilities with water and soap and diarrhea episodes among children U5 in Lao PDR using data from the LSIS II. We have long conducted research concerning community health in rural Lao PDR. The present study was conducted as a part of a larger study that aims to inform strategies to promote hand-washing with soap in rural Lao PDR.

## 2. Materials and Methods

### 2.1. Source of Data and Sample

The present study used data from the LSIS II that were obtained from UNICEF (https://mics.unicef.org/surveys). Out of 11,812 children U5 eligible for the LSIS II, data were available for 11,720 children whose primary caretakers responded to the survey. From the 11,720 children, the present study excluded 316 children because of missing values or “don’t know” responses to question items that were relevant to the present study. Overall, the present study included 11,404 children U5 with 9038 primary caretakers in 8640 households (spreadsheet S1: Dataset). 

### 2.2. Outcome Variable

The outcome variable was the presence of diarrhea illness episodes in children U5 in the two weeks preceding the survey. The outcome was measured by asking the following question to primary caretakers: “In the last two weeks, has (child name) had diarrhea?”

### 2.3. Main Predictor Variable

The predictor variable of interest was the presence of hand-washing facilities with water and soap. This variable had four categories: (1) hand-washing facilities with both water and soap, (2) hand-washing facilities with water alone, (3) hand-washing facilities without water and with/without soap, and (4) no hand-washing facilities. The variable was developed on the basis of three original variables of the LSIS II: “hand-washing facility,” “water availability,” and “soap availability.” “Hand-washing facility” was measured by the following question item: “Can you please show me where members of your household most often wash their hands?” “Water availability” was measured by the following observation item: observe the presence of water at the place for hand-washing. “Soap availability” was measured by the following observation item: is soap, detergent, or ash/mud/sand present at the place for hand-washing?

### 2.4. Other Predictor Variables

Based on the LSIS II report and similar studies that examined the factors associated with childhood diarrhea, the present study included the 12 predictor variables below, which were categorized into four levels: the individual, caretaker, household, and village levels [16,17,18,19,20].

The individual-level variables included the sex and age of the child U5 and the supervision by primary caretaker (adequate/inadequate). The caretaker-level variables included the age and educational attainment of the primary caretaker (no formal education, early childhood education/primary/lower secondary, or above). The household-level variables included the number of household members (six or fewer/over six), sanitation facilities (improved/unimproved/no facilities), source of drinking water (improved/unimproved), ownership of domestic animals (yes/no), and household wealth (quartiles). To assess household wealth, an asset-based index was originally built using principal component analysis [21]. The main predictor variable was also included in the household-level variables. The village-level variables included living area (urban/rural) and source water quality determined by the level of *Escherichia coli* (*E. coli*) contamination in the three household-based samples per village (<11 colony-forming units (CFUs) in all three samples/≥11 CFUs in one or two samples/≥11 CFUs in all three samples). Table A1 (Appendix B) presents detailed explanations of these predictor variables.

### 2.5. Statistical Analysis

Bivariate analyses were conducted to assess the association between the outcome variable and each of the predictor variables using Fisher’s exact test or the chi-square test. Multivariate analyses were conducted using mixed-effect logistic regression. In the multivariate analyses, multi-level modeling was used to account for the hierarchical structure of the data: individuals (level 1) were nested within caretakers (level 2), caretakers were nested within households (level 3), and households were nested within villages (level 4). In the multivariate analyses, three models were examined: in Model 1, only the predictor of interest was included. In Model 2, the predictor of interest and the household-level variables were included. In Model 3, all of the predictor variables were included. The likelihood ratio test was used to assess the fitness of these models. The significance level was set at *p* < 0.05 for all tests. The presence of multicollinearity was assessed using variance inflation factor (VIF) and a VIF > 5 was considered to indicate multicollinearity. These analyses were performed using Stata 14.2 (Stata Corp LP, College Station, TX, USA).

## 3. Results

### 3.1. Characteristics of the Study Participants

Of the 11,404 children U5, 803 (7.0%) children had experienced a diarrhea episode in the two weeks preceding the survey (Table 1). Approximately half (*n* = 5805, 50.9%) of the children U5 were male. The number of children in each age group did not differ greatly, ranging from 2187 (19.2%) in 1 year to 2413 (21.2%) in 3 years. The majority (*n* = 9990, 87.6%) of the children U5 were supervised adequately by their primary caretaker.

### 3.2. Characteristics of the Study Participants’ Households

The median age of the primary caretakers (interquartile range) was 28 years (range = 23–33). Of the 9038 primary caretakers, 7148 (79.1%) had completed at least primary education (Table 2). The median number of household members (interquartile range) was six (range = 5–7). Among the 8640 households with at least one child U5, hand-washing facilities were observed in 7815 (90.5%) households, water was available in the hand-washing facilities of 7235 (83.7%) households, and soap was available in the hand-washing facilities of 4279 (49.5%) households. Nearly half of the households (*n* = 4241, 49.1%) possessed hand-washing facilities with both water and soap available, whereas 2994 (34.7%) households possessed hand-washing facilities with water alone. Meanwhile, 580 (6.7%) households possessed hand-washing facilities without water and with/without soap, and 825 (9.5%) households did not possess any hand-washing facilities. The majority (*n* = 6062, 70.2%) of the households possessed improved sanitation facilities, while 2350 (27.2%) households did not have any sanitation facilities. Most households had an improved source of drinking water (*n* = 7106, 82.2%) and owned domestic animals (*n* = 7122, 82.4%). The number of households in each household wealth quintile was almost the same across quintiles, ranging from 2141 (24.8%) in the fourth group (richest group) to 2176 (25.2%) in the third group (second richest group).

### 3.3. Characteristics of the Study Participants’ Villages

Of the 1159 villages included in the LSIS II, 792 (68.3%) were rural villages (Table 3). In terms of *E. coli* contamination, the quality of source water was considered to be safe in 159 villages (13.7%), whereas it was considered to be unsafe in 419 villages (36.2%).

### 3.4. Bivariate Analyses

The factors significantly associated with diarrhea episodes among children U5 were the sex of the child, the age of the child, the supervision by the primary caretaker, the age of the primary caretaker, the educational attainment of the primary caretaker, the number of household members, soap availability, hand-washing facilities with water and soap, sanitation facilities, source of drinking water, and household wealth (Table 4). No significant differences were found for the rest of the variables.

### 3.5. Multivariate Analyses

In Model 1, the children whose households possess hand-washing facilities with water alone were significantly more likely to have a diarrhea episode compared to the reference group (i.e., the children whose households possess hand-washing facilities with both water and soap) (Table 5). This difference remained significant even after adjusting for the other predictor variables in Models 2 and 3. Children whose households possess hand-washing facilities without water and with/without soap were significantly more likely to have a diarrhea episode compared to the reference group in Models 1 and 2. However, the association became insignificant in Model 3. There were no significant differences in the odds ratio of diarrhea episodes between children whose households do not possess hand-washing facilities and the reference group.

Additionally, in Model 3 there was a significant difference between the reference group and comparison group in the following characteristics; sex of child, age of child, supervision by primary caretaker, sanitation facilities, and household wealth.

## 4. Discussion

The main finding of the present study was that children whose households possess hand-washing facilities with both water and soap were significantly less likely to experience diarrhea episodes compared to children whose households possess hand-washing facilities with water alone. This finding suggests that in the Lao setting, hand-washing with soap is more effective for preventing childhood diarrhea episodes compared to hand-washing without soap. This finding is important because hand-washing facilities with water are available in most households in Lao PDR. If soap use becomes more common, the mortality and morbidity due to diarrhea could be widely reduced.

This main finding is biologically plausible. Analysis of the samples collected at Lao healthcare facilities showed that the major etiologic agents of acute childhood diarrhea are rotavirus, *Escherichia coli*, and *Salmonella* spp. [22], which are transmitted from person-to-person via contaminated hands in households. A community-based study involving 1159 households in rural Lao PDR showed that enteropathogen infections are strongly correlated within members of the same household, suggesting the importance of intra-household transmission [14]. A randomized controlled trial with volunteers in the U.K. showed that hand-washing with plain soap is more effective for the removal of bacterial pathogens from hands than hand-washing with water alone [23]. A community-based randomized control trial with mothers in Bangladesh also showed that hand-washing with a bar of soap is more effective for reducing the bacterial load of coliforms and *Clostridium perfringens* compared to hand-washing with water alone [24]. Additionally, an experimental study with volunteers in the U.S. showed that hand-washing with hand soap and water is effective for reducing viral contamination from finger pads [25].

The main finding is also consistent with those reported from similar observational studies. A cross-sectional study involving 347 households in rural Bangladesh showed that children U5 whose family members washed their hands with soap after defecation were significantly less likely to experience a diarrhea episode in the 48 h preceding the survey compared to children whose family members washed their hands with water only [26]. A cross-sectional study in Malawi, which used Demographic and Health Survey data, showed that the lack of soap in hand-washing facilities was associated with higher odds of having a diarrhea episode among children U5 [27]. In contrast, the main finding of the present study is not consistent with the findings of a study conducted in Saravan province of the Lao PDR. There are two possible reasons for this inconsistency: first, the Saravan study used all household member’s diarrhea episodes as the outcome, suggesting that the reason for the discrepancy could be due to methodological differences. Second, the Saravan study included only 46 diarrhea cases as the outcome, suggesting that the study likely suffered from type II errors; i.e., false negatives.

Although the effect of soap being present in hand-washing facilities on diarrhea incidence was not large (i.e., 5.9% among children in households with soap vs. 8.1% among children in households without soap), placing soap in hand-washing facilities could widely impact the health of Lao children, as more than one-third of Lao households do not have soap in their hand-washing facilities. Based on the assumption that a household has one child U5, 283,000 out of the total 786,000 children U5 in Lao could benefit from placing soap in hand-washing facilities. Additionally, the use of soap could contribute to preventing not only diarrhea, but also other illnesses including pneumonia, which is also a leading cause of death among Lao children [6,28].

The reasons for the absence of soap in hand-washing facilities in many households remain poorly understood in Lao PDR, as no study has been conducted in the country to explore these reasons. The LSIS II report showed, however, that there are some household trends for the absence of soap in handwashing facilities: rural households, households whose heads have lower educational attainment, households of lower wealth quintiles, and households of minority language groups are less likely to have soap in their hand-washing facilities compared to their counterparts [4]. A study on hand-washing facilities in 51 countries reported similar trends: universally, households of higher wealth quintiles and urban households are more likely to have soap in their hand-washing facilities, compared to their counterparts [29]. In Lao PDR, however, soap seems to be affordable for many people: the average price of a bar of soap was 3110 kip (approximately 0.34 U.S. dollars) in 2017 [30]. Considering these trends and the price of soap, further study is necessary to identify the barriers to placing soap in hand-washing facilities in Lao PDR.

The results of the present study also showed that there was no significant difference in the incidence of diarrhea between households with hand-washing facilities where soap and water are available and households without hand-washing facilities. A possible explanation for the lack of a difference is that the households without hand-washing facilities include a substantial proportion of households that live near a community well, and thus household members use the community well as a hand-washing facility. A community-shared well is commonly seen in rural villages of Lao PDR. In fact, wells are a major source of water for housework, including hand-washing, in rural Lao PDR [4]. Additionally, according to the LSIS II survey, in the 66.7% of households their toilet facilities were located not in their houses but in their yards. Thus, there is a possibility that household members have little difficulty in using a community well after defecation, if they live near a community well.

Likewise, in the present study there was no significant difference in the incidence of diarrhea between households with improved sanitation facility and households with no sanitation facility. Currently, we are unable to provide a possible reason for the lack of a difference. A multi-country case control study, which assessed sanitation and hygiene-specific risk factors for moderate-to-severe childhood diarrhea, also showed that there was no significant difference in the risk of diarrhea between households with private sanitation facilities and households with no sanitation facility in, for example, Bangladesh [31]. However, the case-control study did not provide any possible reasons.

A major limitation of the present study is the absence of information about actual hand-washing practices. It is of concern whether the study participants of the households where soap is available in hand-washing facilities actually use soap, as studies have shown that in settings where soap is available, people do not necessarily use soap when washing their hands before/after critical events such as fecal contact, food preparation, eating, and feeding a child. A school-based study in Lao PDR observed that of the pupils who used the school toilet, only 23.9% washed their hands with soap afterward [12]. A multi-country study that evaluated the validity of rapid observation measures of hand-washing practices concluded that the observation of hand-washing materials in hand-washing facilities is a valid measure of hand-washing with soap, although the use of soap is often suboptimal: 27–82% of the primary caretakers of children U5 used soap after possible fecal contact, and overall, they used soap before 24–36% of critical events [32].

Another limitation is that the present study was not able to incorporate all the factors which are reported to be associated with childhood diarrhea episodes in similar studies. Such factors include food preparation practices and child feces disposal practices. For example, a cross-sectional study in Viet Nam showed that the risk of childhood diarrhea was significantly higher among children whose mothers prepared food for cooking somewhere other than the table, compared to children whose mothers prepared food on the table [33]. A cross-sectional study using the data of the 2013 Nigerian Demographic and Health Survey reported that the increased risk of childhood diarrhea was significantly associated with unsafe child feces disposal practices of caretakers [16].

In order to maximize the effect of hand-washing on preventing communicable diseases, merely recommending hand-washing with soap before/after critical events is not enough. The Centers for Disease Control and Prevention recommends the five steps for domestic hand-washing: wetting hands with clean, running water; lathering hands by rubbing hands together with soap; scrubbing hands for at least 20 s; rinsing hands well under clean, running water; and drying hands using a clean towel or air dry hands [34]. A community-based study with primary caregivers of school children in Zimbabwe demonstrated the importance of these five steps in removing microbial contamination [35]. A health education intervention study in Hong Kong showed that the five-steps hand hygiene technique was effective in reducing the spread of infectious diseases in the special education school setting [36]. Because children learn hand-washing from their primary caretakers whose hand hygiene practices are sometimes suboptimal [37,38], health education interventions on hand hygiene to children and caretakers are recommended to promote effective hand-washing.

The results of the present study showed that most of the households with children U5 already had improved sanitation facilities (70.2%) and improved source of drinking water (82.2%), whereas only 49.1% of the households with children U5 had hand-washing facilities with water and soap. The results suggest that hygiene education does not keep up with the increased coverage of sanitation and water supply. Therefore, more efforts should be made in promoting hygiene education in Lao PDR. The proportion of the population that use hand-washing facilities with water and soap is one of the indicators for Target 6.2 of Sustainable Development Goal 6: “By 2030, achieve access to adequate and equitable sanitation and hygiene for all and end open defecation, paying special attention to the needs of women and girls and those in vulnerable situations” [39]. The present study showed that the proportion of households using hand-washing facilities with water and soap is 49.1%, suggesting that continued efforts are needed to achieve Target 6.2 in Lao PDR. In 2017, globally, 60% of the population had basic hand-washing facilities with water and soap, whereas 22% had limited hand-washing facilities lacking water and/or soap [40]. Therefore, many countries, including Lao PDR, face the same challenge: trying to increase the population using basic hand-washing facilities with water and soap.

## 5. Conclusions

The absence of soap in hand-washing facilities was associated with higher odds of having a diarrhea episode among children under five years of age. This suggests that hand-washing with soap is effective for preventing childhood diarrhea in Lao household settings. Households can reduce the risk of diarrhea among their children by making soap available in hand-washing facilities. Further study is necessary to inform strategies for increasing the availability of soap in hand-washing facilities in every household of Lao PDR.

## Figures and Tables

**Table 1 ijerph-18-00687-t001:** Characteristics of the study participants (*n* = 11,404).

Characteristics	*n*	%
Diarrhea episode		
No	10,601	93.0
Yes	803	7.0
Sex of child		
Male	5805	50.9
Female	5599	49.1
Age of child		
0 year	2247	19.7
1 year	2187	19.2
2 years	2317	20.3
3 years	2413	21.2
4 years	2240	19.6
Supervision by primary caretaker		
Adequate	9990	87.6
Inadequate	1414	12.4

**Table 2 ijerph-18-00687-t002:** Characteristics of the study participants’ households (*n* = 8640).

Characteristics	*n*	%
Age of primary caretaker (*n* = 9038)		
<20 years	653	7.2
20–29 years	4632	51.3
30–39 years	2792	30.9
≥40 years	961	10.6
Educational attainment of primary caretaker (*n* = 9038)		
No formal education/early childhood education	1890	20.9
Primary	3543	39.2
Lower secondary or above	3605	39.9
Number of household members		
≤6 people	5959	69.0
>6 people	2681	31.0
Hand-washing facilities		
Yes	7815	90.5
No	825	9.5
Water availability		
Yes	7235	83.7
No	1405	16.3
Soap availability		
Yes	4279	49.5
No	4361	50.5
Hand-washing facilities with water and soap ^1^		
Facility (+), water (+), soap (+)	4241	49.1
Facility (+), water (+), soap (−)	2994	34.7
Facility (+), water (−), soap (+/−)	580	6.7
Facility (−), water (−), soap (−)	825	9.5
Sanitation facilities		
Improved sanitation facilities	6062	70.2
Unimproved sanitation facilities	228	2.6
No facilities	2350	27.2
Source of drinking water		
Improved	7106	82.2
Unimproved	1534	17.8
Ownership of domestic animals		
Yes	7122	82.4
No	1518	17.6
Household wealth		
First (poorest)	2166	25.1
Second	2157	25.0
Third	2176	25.2
Fourth (richest)	2141	24.8

^1^ (+) indicates presence, whereas (–) indicates absence.

**Table 3 ijerph-18-00687-t003:** Characteristics of the study participants’ villages (*n* = 1159).

Characteristics	*n*	%
Area		
Urban	367	31.7
Rural	792	68.3
Source water quality		
<11 CFUs in all three samples	159	13.7
≥11 CFUs in one or two samples	581	50.1
≥11 CFUs in all three samples	419	36.2

CFUs, colony-forming units.

**Table 4 ijerph-18-00687-t004:** Bivariate analyses of the factors associated with diarrhea episodes (*n* = 11,404).

Characteristics	No Diarrhea Episode		Diarrhea Episode		*p*-Value ^1^
	*n*	%	*n*	%	
Sex of child					
Male	5364	92.4	441	7.6	0.019
Female	5237	93.5	362	6.5	
Age of child ^2^					
0 year	2039	90.7	208	9.3	<0.001
1 year	1964	89.8	223	10.2	
2 years	2162	93.3	155	6.7	
3 years	2299	95.3	114	4.7	
4 years	2137	95.4	103	4.6	
Supervision by primary caretaker					
Adequate	9331	93.4	659	6.6	<0.001
Inadequate	1270	89.8	144	10.2	
Age of primary caretaker					
<20 years	713	90.0	79	10.0	0.002
20–29 years	5738	92.8	443	7.2	
30–39 years	3165	93.9	204	6.1	
≥40 years	985	92.7	77	7.3	
Educational attainment of primary caretaker					
No formal education/early childhood education	2354	91.4	221	8.6	0.003
Primary	4148	93.4	291	6.6	
Lower secondary or above	4099	93.4	291	6.6	
Number of household members					
≤6 people	6712	93.4	473	6.6	0.014
>6 people	3889	92.2	330	7.8	
Hand-washing facilities					
Yes	9577	92.9	729	7.1	0.756
No	1024	93.3	74	6.7	
Water availability					
Yes	8838	93.1	651	6.9	0.096
No	1763	92.1	152	7.9	
Soap availability					
Yes	5109	94.1	323	5.9	<0.001
No	5492	92.0	480	8.0	
Hand-washing facilities with water and soap ^3^					
Facility (+), water (+), soap (+)	5068	94.1	319	5.9	<0.001
Facility (+), water (+), soap (−)	3770	91.9	332	8.1	
Facility (+), water (−), soap (+/−)	739	90.5	78	9.5	
Facility (−), water (−), soap (−)	1024	93.3	74	6.7	
Sanitation facilities					
Improved sanitation facilities	7289	93.6	500	6.4	<0.001
Unimproved sanitation facilities	266	87.2	39	12.8	
No facilities	3046	92.0	264	8.0	
Source of drinking water					
Improved	8619	93.2	624	6.8	0.013
Unimproved	1982	91.7	179	8.3	
Ownership of domestic animals					
Yes	8754	92.9	669	7.1	0.629
No	1847	93.2	134	6.8	
Household wealth					
First (poorest)	2689	91.5	250	8.5	<0.001
Second	2560	92.2	216	7.8	
Third	2656	93.3	190	6.7	
Forth (richest)	2696	94.8	147	5.2	
Area					
Urban	2856	93.6	195	6.4	0.107
Rural	7745	92.7	608	7.3	
Source water quality					
<11 CFUs in all three samples	1439	93.6	98	6.4	0.094
≥11 CFUs in one or two samples	5084	92.4	417	7.6	
≥11 CFUs in all three samples	4078	93.4	288	6.6	

^1^ Fisher’s exact test. ^2^ Chi-square test. ^3^ (+) indicates presence, whereas (−) indicates absence.

**Table 5 ijerph-18-00687-t005:** Multivariate analysis of the association between diarrhea episodes among children under five and hand-washing facilities (*n* = 11404).

Characteristics	Model 1		Model 2		Model 3	
	OR	95% CI	AOR	95% CI	AOR	95% CI
Sex of child						
Male					1.00	Reference
Female					0.82	0.69–0.98
Age of child						
0 year					1.00	Reference
1 year					1.14	0.90–1.46
2 years					0.63	0.49–0.82
3 years					0.41	0.31–0.54
4 years					0.40	0.30–0.54
Supervision by primary caretaker						
Adequate					1.00	Reference
Inadequate					1.72	1.35–2.20
Age of primary caretakere						
<20 years					1.00	Reference
20–29 years					0.88	0.64–1.22
30–39 years					0.78	0.55–1.11
≥40 years					1.15	0.75–1.77
Educational attainment of primary caretaker						
No formal education/early childhood education					1.00	Reference
Primary					0.82	0.64–1.04
Lower secondary or above					0.97	0.73–1.29
Number of household members						
≤6 people			1.00	Reference	1.00	Reference
>6 people			1.17	0.97–1.40	1.12	0.92–1.35
Hand-washing facilities with water and soap ^1^						
Facility (+), water (+), soap (+)	1.00	Reference	1.00	Reference	1.00	Reference
Facility (+), water (+), soap (−)	1.49	1.22–1.81	1.28	1.03–1.58	1.31	1.05–1.63
Facility (+), water (−), soap (+/−)	1.71	1.22–2.39	1.50	1.07–2.12	1.41	0.99–2.02
Facility (−), water (−), soap (−)	1.18	0.86–1.63	1.05	0.76–1.46	1.03	0.74–1.45
Sanitation facilities						
Improved sanitation facilities			1.00	Reference	1.00	Reference
Unimproved sanitation facilities			2.14	1.35–3.37	2.07	1.29–3.32
No facilities			0.99	0.78–1.25	1.01	0.79–1.29
Source of drinking water						
Improved			1.00	Reference	1.00	Reference
Unimproved			1.11	0.88–1.41	1.13	0.89–1.45
Ownership of domestic animals						
Yes			1.00	Reference	1.00	Reference
No			1.03	0.81–1.30	1.02	0.80–1.30
Household wealth						
First (poorest)			1.00	Reference	1.00	Reference
Second			0.92	0.72–1.17	0.90	0.70–1.16
Third			0.82	0.62–1.09	0.80	0.59–1.08
Forth (richest)			0.66	0.48–0.91	0.62	0.43–0.88
Area						
Urban					1.00	Reference
Rural					0.90	0.69–1.18
Source water quality						
<11 CFUs in the all the samples					1.00	Reference
≥11 CFUs in one or two samples					1.22	0.88–1.68
≥11 CFUs in all the samples					0.99	0.70–1.38
Log likelihood		−2846.48		−2834.31		−2767.01
Likelihood ratio test (*p*-value)		0.0002		0.0020		<0.0001

^1^ (+) indicates the presence, whereas (−) indicates the absence. OR, odds ratio; CI, confidence interval; AOR, adjusted odds ratio.

## Data Availability

The data are available at the Appendix A.

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
