# Peer review of "Effects of Hand-Washing Facilities with Water and Soap on Diarrhea Incidence among Children under Five Years in Lao People’s Democratic Republic: A Cross-Sectional Study"

_ijerph, 2021, doi:10.3390/ijerph18020687_

Round 1

Reviewer 1 Report

Title: Effects of Hand Washing Facilities with Water and Soap on Diarrhea Incidence Among Children Under Five Years in Lao People’s Democratic Republic: A Cross-Sectional Study.

This study assessed the association between the presence of household 14 hand washing facilities with water and soap and diarrhea episodes among children U5 in Lao PDR.

The study has relevance to child public health, communicable disease transmission and hand hygiene behaviour. Diarrhoeal disease remains the largest killer of young children under the age of five years old, particularly in developing countries. Adequate hand hygiene practice and compliance is essential in preventing the transmission of diarrhoea amongst other diseases. The provision of adequate hand hygiene amenities and facilities is equally important; and the authors of this study have chosen to focus on this factor as there is little research in this area.

Overall the paper presented is well written, however there is room for improvement:

In sections 2.3 and 2.4, while there is mention of the main predictor and alternative variables which determined the methodological approach adopted, one additional factor which may also contribute to the incidence rate of diarrhoeal disease not mentioned is food preparation practices. Although this may be difficult to factor in, it may be worth mentioning as a potential limiting factor.

While the results suggest that the provision of water and soap was associated with lower incidence of diarrhoeal disease amongst young children in those households versus those with water alone, the researchers should also mention in the discussion that the lower incidence rate could also be due to the handwashing technique being taught to children by role models (caretakers), which has been suggested in other studies as being significant in determining hand hygiene behaviour and communicable disease transmission rates:

  • Okyay, P., Ertug, S., Gultekin, B., Onen, O., Beser, E. 2004. Intestinal parasites prevalence and related factors in school children, a western city sample-Turkey. BMC Public Health.
  • Parveen, S., Nasreen, S., Allen, J.V., (...), Luby, S.P., Ram, P.K. 2018. Barriers to and motivators of handwashing behavior among mothers of neonates in rural Bangladesh. BMC Public Health, Vol. 18(1), pp. 483.

Equally, other studies have suggested that differences between washing with water alone, and washing with soap are not significant in reducing communicable disease prevalence – suggesting that other factors like hand hygiene technique and in particular the number of handwashing steps that are followed and how long people spend washing hands are much more significant. The following papers highlight this and it may be worth considering and acknowledging in the discussion:

  • Choi, K.S., P.K. Wong, P.K., W.Y. Chung, W.Y. 2012. Using computer-assisted method to teach children with intellectual disabilities handwashing skills. Disabil Rehabil Assist Technol, 7, pp. 507-516.
  • Lee, R.L.T., Leung, C., Tong, W.K., Chen, H., Lee, P.H. 2015. Comparative efficacy of a simplified handwashing program for improvement in hand hygiene and reduction of school absenteeism among children with intellectual disability. American Journal of Infection Control. Vol. 43(9), pp. 907-912.
  • Friedrich, M.N.D., Julian, T.R., Kappler, A., Nhiwatiwa, T., Moslera,H.J. 2017. Handwashing, but how? Microbial effectiveness of existing handwashing practices in high-density suburbs of Harare, Zimbabwe. American Journal of Infection Control. Vol. 45(3), pp. 228-233.

There were no apparent spelling or grammatical errors in this paper. Although in Lines 309 and 338 the numbers need changed from red colour to black before publishing.

Author Response

Thank you so much for having reviewed our manuscript and provided valuable comments. Our responses to your comments were shown in the file attachment.

Reviewer 2 Report

For a start, I think that you have to provide a background. What brought your attention to Laos, and why did think that it would be of specific interest to study the benefits of handwashing over there? Or why did you suspect that handwashing would be less beneficial in Laos than in any other country?

Did you have any doubts of the merits of good hand hygiene in general? As you may know, Ignaz Semmelweis established the life-saving benefits of handwashing in the mid 19th century, However, it was not until 10+ years later that his finding found general acceptance. Within Public Health, the general perception is that what Semmelweis had discovered; it is something that still holds true today, and few public health workers challenge that hand-washing is one of the most important tools in public health. Concerning U5-children, there is a wide consensus that it can keep kids from getting the flu, prevent the spread of disease and keep infections at bay.

Observe that it is referred to as one of the most important tools. It is also important with regular handwashing routines. The observation of washing facilities without water indicates that water supply may be intermittent, thereby compromizing the benefits of handwashing. Some of your references mention other causes of diarrhea, but you make no use of that information. That’s a shame. Overall, I think that you treat your references in a very summary way. Reference 14 deals with a study, which is based on an approach which is very similar to the one you use. Is that where you got your inspiration from?

Your statistical calculations are very much of a standard type. They do not represent a significant scientific feat. The results need some in-depth comments to become interesting. To me, it does not seem likely that people take off to the community well for handwashing after relieving themselves.

Drinking water quality seems to have a minor impact on the number of diarrhea episodes. Concerning santiation, there are some interesting relationships. The worst alternative seems to be to have access to unimproved sanitation facilities - it is actually worse than having no sanitation facility. To me, this something that really needs to be commented on.

For a long time,water engineers believed that drinking water supply was the key to better health. Later, they found out that water without sanitation is not enough. During the current century, there has been a growing understanding that we need a third link - hygiene. Thus, there are now innumerable water, sanitation and health programs running in developing countries. When they are well run, they are very beneficial, but there are still too many of them that are poorly run. Anyhow, I am surprised that you do not discuss this movement at all.

Author Response

(The authors gave the same response as above.)

Reviewer 3 Report

The work is a clear description about diarrhea case among children under 5 in different cases. The alternative cases have been described well and analyzed statistically in order to show alternative factors which increase the risk of diarrhea.

The results are explained well. Hand washing using both water and soap is shown to be the best way to avoid diarrhea and hand-washing using only water is less effective. Washing is an important factor and already children at age 3-4 years can wash or co-work in hand-washing – and maybe they are not as exposed to get diarrhea as smaller children. In addition, improved sanitation reduced diarrhea.

The conclusions are clear.  In my country soap has been done by cooking ash lime and animal fat.

Author Response

(The authors gave the same response as above.)

Reviewer 4 Report

The present study was based on establishing associations between the presence of household hand washing facilities with water and soap and diarrhea episodes among children under 5 years of age in Lao PDR. Although it is a study of a very specific population and only one aspect is studied, it is an interesting work. Only small aspects need to be checked.

According to the structure presented in the results, the wording of the results should be improved, since not all the information that appears in the text (e.g. Age of child, ≥11 CFUs in one or two samples…). Please, this should be carefully checked for all variables.

Specific comments:

Line 47-48. I suggest modifying the wording and joining the two sentences, eliminating the second time “rapid observations” appears written.

Line 153 (3.4. Bivariate analyses). I consider the information presented in the text to be redundant, being able to eliminate the p-value of the text, since it duplicates that presented in the tables. I would also add a sentence that indicates something like: No differences were found for the rest of the variables.

Table 4. Only the chi-square test was used for the variable Age of child? And for Age of primary caretaker?, and in some other variable?

The link provided in Reference 30 is not available.

Author Response

(The authors gave the same response as above.)

Round 2

Reviewer 2 Report

It seems rather strange to me that you do not at all refer to anything that you have found in the studies that you say that you have undertaken. That reduces your paper to a statistical manipulation of the data of an official survey, which was undertaken with a focus on a different purpose. For the statistics, you just use existing software. To explain the results, you essentially use findings from international studies, although your aim was to find if there was anything peculiar about handwashing in Laos. I am not familiar with Laos, but given the international quest for WASH-projects, I am rather convinced that there have a fair number of such projects in Laos. It is a major deficiency of your paper that it does not mention anything about these. The paper needs to be put in context As it stands, I find it to be too thin for publication in a scientific journal.

Author Response

(The authors gave the same response as above.)
